# Quantitative Analysis of Soil Cd Content Based on the Fusion of Vis-NIR and XRF Spectral Data in the Impacted Area of a Metallurgical Slag Site in Gejiu, Yunnan

**Zhenlong Zhang [1], Zhe Wang [1,\*], Ying Luo [1], Jiaqian Zhang [1], Xiyang Feng [1], Qiuping Zeng [1], Duan Tian [1], Chao Li [1], Yongde Zhang [1], Yuping Wang [2], Shu Chen [1] and Li Chen [3]**

[1] College of Environment and Resources, Southwest University of Science & Technology, Mianyang 621010, China

[2] Division of International Applied Technology, Yibin University, Yibin 644000, China

[3] College of Natural Resources and Environment, Northwest A&F University, Xianyang 712100, China

\* Correspondence: wz2004@126.com

**Abstract:** Vis-NIR and XRF spectroscopy are widely used in monitoring heavy metals in soil due to their advantages of being fast, non-destructive, cost-effective, and non-polluting. However, when used individually, XRF and vis-NIR may not meet the accuracy requirements for Cd determination. In this study, we focused on the impact area of a non-ferrous metal smelting slag site in Gejiu City, Yunnan Province, fused the pre-selected vis-NIR and XRF spectra using the Pearson correlation coefficient (PCC), and identified the characteristic spectra using the competitive adaptive reweighted sampling (CARS) method. Based on this, a quantitative model for soil Cd concentration was established using partial least squares regression (PLSR). The results showed that among the four fusion spectral quantitative models constructed, the model combining vis-NIR spectral second-order derivative transformation and XRF spectral first-order derivative transformation (D2(vis-NIR) + D1(XRF)) had the highest coefficient of determination ($R^2 = 0.9505$) and the smallest root mean square error (RMSE = 0.1174). Compared to the estimation models built using vis-NIR and XRF spectra alone, the average computational time of the fusion models was reduced by 68.19% and 63.92%, respectively. This study provides important technical means for real-time and large-scale on-site rapid estimation of Cd content using multi-source spectral fusion.

**Keywords:** soil Cd pollution; visible-near infrared; X-ray fluorescence; spectral fusion; competitive adaptive reweighted sampling

## 1. Introduction

Heavy metal pollution in soil has become a global environmental issue [1]. In particular, due to human activities such as mining, industrial waste discharge, and improper use of pesticides, land degradation continues to worsen [2]. According to a national survey, the concentration of Cd in soil was 7.0%, which is significantly higher than that of other heavy metals [3]. Cd could further enter the human body through the food chain, posing a serious threat to human health [4,5]. Indeed, timely determination of Cd concentrations in soil can prevent the spread of Cd and provide a reference for the management of Cd pollution in environmental systems. Traditional methods for determining the concentration of Cd in soil still primarily rely on geochemical methods [6], but their disadvantages, such as long analytical cycles, complicated sample preparation, high analytical cost, and impracticality of on-site monitoring, are not suitable for this study [7]. Therefore, it is necessary to establish an accurate, stable, and rapid method for the determination of soil Cd concentration and to complement it with the integration of multi-source spectral data.

The visible-near infrared (vis-NIR) technology possesses advantages such as time-saving, effectiveness, affordability, and environmental friendliness [8]. Vis-NIR spectroscopy has shown its maturity in predicting cadmium-related soil physicochemical



properties, such as iron oxides and organic matter [9]. Previous studies have indicated that vis-NIR spectra can indirectly measure the Cd content in soil [10]. The Chinese standard for risk control of soil pollution on agricultural land stipulates (GB 15618-2018) that the screening limit for Cd in agricultural soil is 0.3 mg/kg [11]. However, the direct measurement of Cd in soil using vis-NIR spectroscopy is limited by the weak signal of low Cd concentrations, affecting the monitoring effect [6,12]. Additionally, the combination of vis-NIR spectroscopy and remote sensing techniques can infer soil heavy metal concentrations at large spatial scales [13]. Portable X-ray fluorescence (pXRF) has been widely applied in heavy metal detection because of its advantages of speed, non-destructiveness, low cost, simultaneous multi-element analysis, and on-site detection [14]. However, the test results of this instrument are highly uncertain due to the influence of soil physicochemical properties, detection limits of elements, and detection principles, making it difficult to predict heavy metal pollution at large spatial scales [15]. Compared to the individual use of vis-NIR or X-ray fluorescence (XRF), the fusion of XRF and NIR spectra can expand the coverage of soil properties [16]. Therefore, the integrated application of proximal soil remote sensing techniques, such as XRF and vis-NIR, for rapid estimation of heavy metal concentrations in soil has been validated [17,18], which can effectively improve prediction accuracy, enhance the monitoring efficiency, and reduce the cost [19]. Among numerous spectral fusion methods, the serial fusion of XRF and vis-NIR spectra has the advantages of simplicity and strong operability [18]. Furthermore, research has shown that combining principal component analysis (PCA) with serial fusion methods can further improve model accuracy [20]. Thus, the utilization of serial XRF and vis-NIR spectra for estimating heavy metal concentrations is a feasible approach. Compared with other fusion methods, feature layer crosstalk is more suitable for rapid estimation of soil Cd content in the field.

The difficulty in predicting Cd concentrations using tandem spectroscopy is the elemental Cd does not have distinct characteristic bands in the vis-NIR spectral region [21]. Moreover, China's low screening limit for Cd in soils further increases the difficulty of identifying characteristic spectral bands in serial spectra. Tan et al. [22] used competitive adaptive reweighted sampling (CARS) for selecting characteristic spectral bands and found that it improved the prediction of arsenic, chromium, lead, and zinc concentrations in soil compared to traditional models. The study suggested that feature spectrum selection plays a key role in enhancing the generalization ability of estimation models. The Pearson correlation coefficient (PCC) has been proven to be statistically based and interpretable for the selected characteristic bands [23]. However, there is limited research on using PCC in conjunction with multisource spectral fusion for estimating Cd content in soil, which can provide technical references for further expanding and optimizing fusion methods. However, it is difficult to address the interrelationships between independent variables by using only PCC to select feature spectra [24]. Estimation models for Cd content built using PCC-selected characteristic spectra may suffer from overfitting, leading to unreliable results. Therefore, it is necessary to combine PCC with other methods to select characteristic spectra [25]. Among these methods, CARS primarily selects feature variables based on the principle of "survival of the fittest", thus removing irrelevant spectral information and retaining variables that effectively enhance the model's adaptability [22,26].

Agricultural land around the impact area of a nonferrous metal processing and smelting slag site in the city of Gejiu, Yunnan Province, has been affected by the nonferrous metal smelting process, and its farmland soil is mainly contaminated by heavy metals such as Cd and Pb [27]. In order to characterize the Cd contamination non-destructively and rapidly in the affected area's soil, a model capable of quantitatively monitoring Cd content is needed to address the limitations of traditional precise analytical monitoring methods in obtaining on-site and rapid measurements of Cd concentration in the study area's soil. Therefore, this study aims to combine XRF and vis-NIR spectra in a serial fusion approach, combined with PCC analysis, to construct a technical roadmap for spectral preprocessing, feature spectrum selection, and quantitative estimation using the PCC_CARS_PLSR (Pearson Correlation Coefficient–Competitive Adaptive Reweighted Sampling–Partial Least Squares Regression)

multi-method approach. This approach enables the rapid assessment of Cd content in the soil, a characteristic pollutant in the agricultural land of the non-ferrous metal selection and smelting slag site in Gejiu City, Yunnan Province. Additionally, $Cd^{2+}$ poses a threat to the natural environment and human health due to its tendency to migrate and diffuse in soil. The detection limit of Cd in XRF spectrometry is lower than that of other heavy metal elements, and the monitoring accuracy is lower than that of other heavy metal elements. In this study, the accuracy of single-spectrum models and spectral fusion models predicting the content of Cd using different transformation methods were compared. This study provides a technical means for rapid estimation of Cd content in the field by multi-source spectral fusion, which lays the foundation for the remediation of Cd-contaminated soil. At the same time, this study also lays a research foundation for future research on dynamic, real-time, and large-scale quantitative monitoring of soil Cd pollution based on hyperspectral remote sensing images.

## 2. Materials and Methods

### 2.1. Study Area

The study area is located within a 5–20 km radius of the non-ferrous metal smelting slag repository in Gejiu City, Honghe Hani, and Yi Autonomous Prefecture, Yunnan Province, China. It covers an approximate area of 6.52 km². The average elevation is around 1300 m, with significant elevation differences. The area has a distinct vertical climate and belongs to a subtropical highland monsoon climate with abundant rainfall. Within the study area, there are 965,935 m² of construction land and 2,609,041 m² of farmland. After preliminary processing, it was found that the soil pH ranged from 5.32 to 8.80, and the soil moisture content ranged from 20% to 40%. The wastewater and sludge generated during non-ferrous metal processing have caused severe heavy metal pollution in the surrounding environment. Cd is one of the most heavily polluted heavy metals in the study area, posing a serious threat to crops and human health [28]. The geographical location of the study area and the geological map are shown in Figures 1 and 2. Most of the study area is underlain by Quaternary soil deposits, except for small portions of dolomite, siltstone, siliceous rock, and chert.

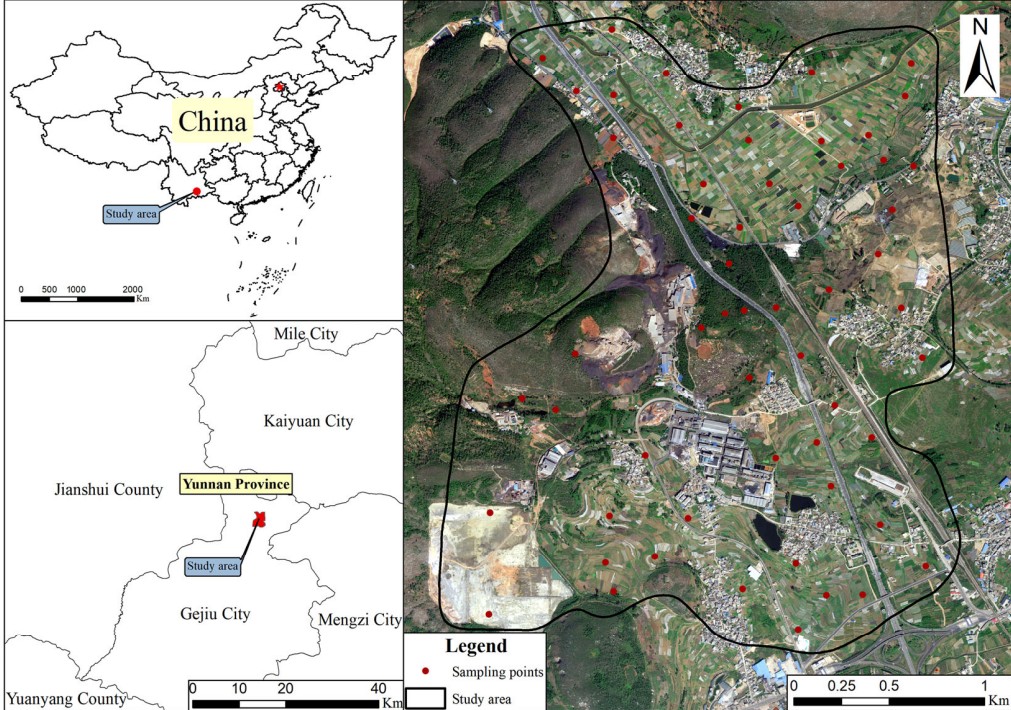

**Figure 1.** Geographical location of the study area.

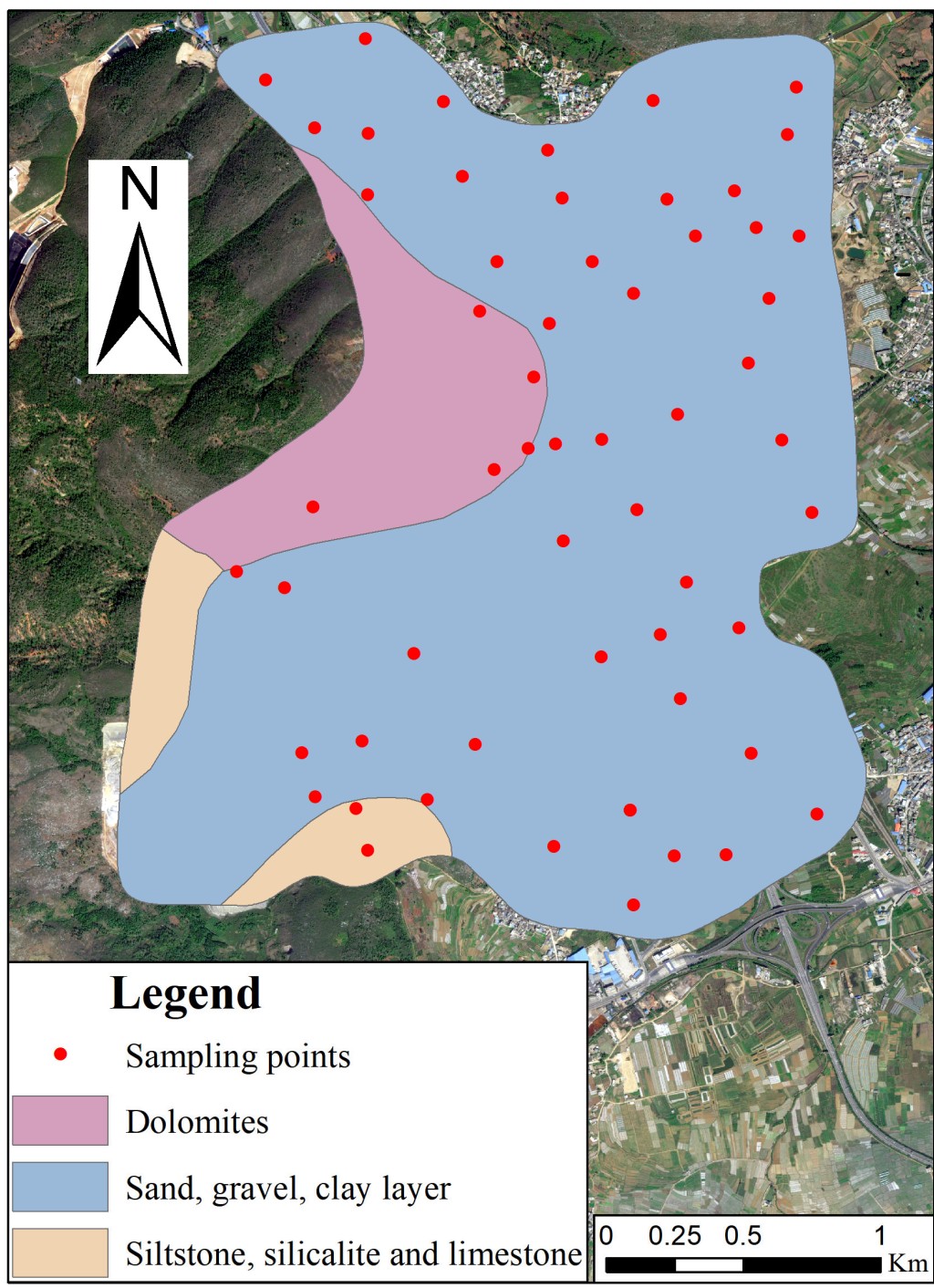

**Figure 2.** Geological map.

*2.2. Data Collection*

2.2.1. Soil Sample Collection

To establish a reliable quantitative estimation model for soil Cd concentration in the vicinity of the slag repository (HJ/T 166-2004), 58 soil samples were collected using a random distribution method. The GPS was used to record the locations of the sampling points. According to the "Technical Specifications for Soil Environmental Monitoring" [29], metal tools should be avoided during soil sample collection. First, the soil surface was cleared of mulch, such as tree branches and weeds. Then, five surface samples (0–5 cm) were taken in a 10 × 10 m area using a diagonal sampling method. The samples were mixed together, discarding impurities such as stones and plant roots. Finally, about 1 kg of

mixed soil samples were sealed in polyethylene plastic bags [30]. The soil samples were dried in the laboratory, then ground and sieved using a 100-mesh nylon sieve (0.150 mm) to remove larger particles. The soil samples were divided into two parts: one for chemical analysis and the other for spectral analysis.

### 2.2.2. Chemical Analysis

The soil samples were digested in a graphite digestion apparatus. First, 0.1 g of the sample was weighed and placed in a Teflon digestion tank. Then, 5 mL of nitric acid ($\rho HNO_3$ = 1.42 g/mL) was added to the vessel to soak the sample for 0.5 h to remove organic matter. After that, 2 mL of hydrofluoric acid ($\rho HF$ = 1.49 g/mL) and 1 mL of perchloric acid ($\rho HClO_4$ = 1.49 g/mL) were added. Finally, the digestion vessel was placed in a graphite digestion block and digested at 180 °C for 4 h. The total Cd concentration in the solution was determined using an Inductively Coupled Plasma Mass Spectrometer (ICP-MS, 800DV, TMO, Waltham, MA, USA) (argon gas source valve pressure reducing valve at about 550 kPa, circulating water pressure indication between 50 and 310 kPa.). Glassware and polyethylene bags were soaked in nitric acid for 12 h before use, and ultrapure water was used to prepare the solutions for analysis [31]. All experimental samples were digested simultaneously with a Cd concentration reference standard in the soil. Sample replicates (approximately 20%) and standard reference materials (GSS-5/GBW07405) [28] were included in each batch of sample digestion and chemical analysis, maintaining quality assurance, quality control, and blank control. The relative standard deviations were less than 5%, and the recoveries of Cd from the standard reference materials were 90–95%, respectively.

### 2.2.3. Measurement of vis-NIR Spectra Using a Portable Spectrometer

Visible-NIR spectra of soil samples were measured in a darkroom using a PSR-2500 portable spectrometer manufactured by Spectrum Evolution (operating instructions are available on the Spectral Evolution website (https://spectralevolution.com/products/software/ (accessed on 4 June 2023))). The soil samples were placed in black containers with a diameter of 10 cm and a depth of 1.5 cm. Prior to measurement, the instrument was preheated for at least 30 min with a 100 W halogen lamp set as the sole light source. Calibration with a white reference panel was performed for optimization. The probe viewing angle was kept at 15°, and the light source incidence angle was 30° during acquisition. The distance between the light source and the center of the soil surface was 50 cm, while the probe was positioned 15 cm above the soil surface. The container was then divided into three directions at an angle of 120°, and five spectra were collected in each direction, resulting in a total of 15 spectra per soil sample. The arithmetic mean of the collected spectra was calculated to obtain the final soil reflectance spectra to facilitate subsequent analysis.

### 2.2.4. Measurement of XRF Spectra Using an X-ray Fluorescence Spectrometer

The XRF spectra of the soil samples were measured using the Niton XL3t 950 XRF spectrometer from Thermo Fisher Scientific, Waltham, MA, USA. After passing through a 200-mesh nylon sieve (0.074 mm), the soil samples were placed in sample cups. The surface of each sample was leveled and covered with a layer of polyester film. Then, the sample cup was placed on the instrument's test stand, and scanning was conducted for 80 seconds per sample. Three scans were performed for each sample, and the average spectrum of the three scans was taken as the result. After removing the low-energy values at the edges, each spectrum retained 3522 data points ranging from 0.39 keV to 53.205 keV. The spectrometer was connected to the computer via a data cable, and the spectral data were exported to the computer using the NDT program, the manual for which is available on the Thermo Fisher Scientific website (https://www.thermofisher.cn/order/catalog/product/10131166?SID=srch-srp-10131166 (accessed on 5 June 2023)).

*2.3. Data Analysis*

In this study, data preprocessing, feature band selection, construction of single-spectrum models, construction of spectral fusion models, and model accuracy evaluation were implemented using the Python 3.10 programming language in PyCharm Community Edition 2022.1.2 software. The specific processing workflow is illustrated in Figure S1.

### 2.3.1. Spectral Pre-Processing Methods

Spectral preprocessing aimed to standardize the format of XRF and vis-NIR spectral data and reduce errors caused by system noise. The spectral data were exported to Microsoft Excel in batches and subjected to wavelet transform (WT) and Savitzky–Golay (SG) smoothing preprocessing. WT applied scaling and translation operations to gradually refine the signal (function) at multiple scales, ultimately achieving automatic focus on the details of the spectral profile [32]. SG smoothing was employed for denoising the spectral curve, and used a second-order polynomial with a window size of 12 for smoothing. Spectral transformations enhanced the spectral signal and increased the correlation between spectral wavelengths and Cd content [33,34]. The raw spectra (RS) were subjected to several transformations, including mean centering transformation (CT), standard normal variate transformation (SNV), multiplicative scatter correction (MSC), first-order derivative transformation (D1), second-order derivative transformation (D2), detrending transformation (DT), continuum removal (CR), and reciprocal logarithm transformation (CL). Due to the presence of zero values in XRF spectra, the CL transformation cannot be applied as the denominator. The transformed results of XRF and vis-NIR spectra are shown in Figures S2 and S3, respectively. Finally, the spectral data were subjected to standardization processing.

### 2.3.2. Feature Spectral Selection

The feature spectral selection process consisted of preliminary screening using the PCC method and subsequent screening using the CARS method.

(1) The correlation analysis was conducted using the PCC method [23]. The correlation coefficients between the Cd concentrations in soil and the spectral bands of both types of spectra were calculated, and the absolute values of the correlation coefficients were obtained. The equation for calculating the correlation coefficient is as follows:

$$r_{(X,Y)} = cov_{(X,Y)} / \sqrt{var(X)} \sqrt{var(Y)} \tag{1}$$

where $r_{(X,Y)}$ is the correlation coefficient between the two variables $X$ and $Y$; $cov_{(X,Y)}$ is the covariance of the two variables; $var(X)$, $var(Y)$ are the variance of the variables.

After conducting multiple experiments, it was found that selecting 200 highly correlated spectral bands from the vis-NIR spectra and 600 highly correlated spectral bands from the XRF spectra ensured model quality and reduced model computation time.

(2) CARS utilized adaptive sampling to retain spectral bands with relatively large absolute coefficients in the PLSR model [35]. Then, a Monte Carlo cross-validation method was employed to model each subset of wavelength variables, and the optimal subset was selected as the feature spectral band set based on the root mean square error of cross-validation (RMSECV).

### 2.3.3. Estimation Model

The Kennard–Stone (KS) algorithm [36] was used to calculate the Euclidean distance of the dataset. The samples corresponding to the maximum and minimum distances were selected as the training set. This process was repeated until the training set reached the specified quantity (80% for training, 20% for validation). Among the 58 soil samples, 46

were selected for the training set, and 12 were selected for the validation set. The formula for calculating the Euclidean distance is as follows:

$$d_x(p, q) = \sqrt{\sum_{j=1}^{J}\left[x_p(j) - x_q(j)\right]^2} \quad p, q \in [1, N] \tag{2}$$

where $J$ is the total number of bands; $N$ is the total number of samples; $x_p(j)$ and $x_q(j)$ are date values at $p$ and $q$ on the $j$ band.

A prediction model was developed using the PLSR method. PLSR is a novel multivariate statistical analysis method that combines the advantages of PCA and linear regression modeling and is more effective in distinguishing between spectral information and noise. Compared with traditional linear models, the most important feature of PLSR is its utilization of data dimensionality reduction and comprehensive information selection techniques. PLSR can model independent variables with multiple correlations, which can improve model accuracy when using spectroscopic techniques to build predictive models for soil Cd content [37].

### 2.3.4. Accuracy and Efficiency Evaluation

The accuracy of the estimation model was evaluated using $R^2$ (coefficient of determination) and RMSE (root mean square error). A higher $R^2$ value indicated better model fit, with values closer to 1 indicating higher accuracy. A smaller RMSE value indicated a better predictive ability for the model. However, it should be noted that the CARS method for feature spectrum selection exhibits randomness [38]. This was reflected when the RMSE of the validation set was greater than that of the training set, indicating that the predictive model had obtained a local solution. In such cases, it is necessary to repeat the variable selection and modeling process until the RMSE of the validation set becomes smaller than that of the training set [39]. The formulas for calculating $R^2$ and RMSE are as follows:

$$R^2 = \sum_{i=1}^{n}(\hat{y}_i - \overline{y_i})^2 \Big/ \sum_{i=1}^{n}(y_i - \overline{y_i})^2 \tag{3}$$

$$\text{RMSE} = \sqrt{\frac{1}{n}\sum_{i=1}^{n}(\hat{y}_i - \overline{y_i})^2} \tag{4}$$

where $\overline{y}$ is the mean value of the sample observations; $\hat{y}$ is the predicted value of the sample; $n$ is the number of samples to be verified.

Due to the relatively slow execution speed of CARS, real-time updating of soil heavy metal Cd concentration is required for rapid on-site estimation. Therefore, the algorithm execution time was considered the efficiency evaluation criterion for different spectral transformation methods in the prediction model evaluation, aiming to distinguish the characteristics of different models in the inversion of Cd concentration.

### 2.3.5. Building a Spectral Fusion Model

First, we conducted a statistical analysis to assess the accuracy and efficiency of the individual spectral models. The models were evaluated based on their $R^2$, RMSE, and computation time on the validation set. The number of models selected varied depending on the different transformation methods applied to the XRF and vis-NIR spectra. Next, the selected models corresponding to the XRF and vis-NIR spectra were concatenated, resulting in a fused spectrum (FS). The FS spectrum contained a total number of spectral bands equal to the sum of the bands in the vis-NIR and XRF spectra. Subsequently, the FS spectrum underwent another round of spectral preprocessing to enhance its quality and minimize potential noise or artifacts. Following that, the PCC_CARS_PLSR method was employed to establish a predictive model for estimating the soil heavy metal Cd content using the fused spectrum. In this modeling process, 600 variables were selected through PCC screening to ensure the inclusion of relevant spectral information contributing to accurate predictions. Finally, the accuracy and efficiency of the FS model were evaluated.

This evaluation involved assessing the accuracy of the model using $R^2$ and evaluating its efficiency by measuring the required training and prediction time. Through this comprehensive evaluation, we gained insights into the predictive accuracy and computational efficiency of the FS model in estimating soil heavy metal Cd content.

## 3. Results

### *3.1. Descriptive Statistical Analysis*

#### 3.1.1. Descriptive Statistical Analysis of Soil Cd Concentration

Descriptive analysis was carried out based on the results of laboratory ICP-MS accurate quantitative analysis tests of Cd content in soil samples from the study area (Table 1). The results of the statistical analysis showed that the mean value of soil Cd was higher than its corresponding median value, indicating a positively skewed mode of normal distribution of Cd in soil. The Cd pollution in the study area was serious, with the average Cd concentration exceeding the background level in Yunnan Province by 30 times, 66 times higher than the risk screening value and 13 times higher than the risk control value in the Chinese Soil Environmental Quality Risk Control Standard for Agricultural Land (GB15618-2018) [11], which poses a serious threat to human health. The standard deviation of soil Cd was 16.3516, indicating significant differences in Cd concentration among different directions in the non-ferrous metal smelting slag depot. The Cd element was influenced by topography, wind direction, and water sources continuously flowing into surrounding areas. The coefficient of variation for Cd concentrations in the study area was 82.25%, suggesting a significant influence of anthropogenic factors on Cd concentrations.

**Table 1.** Descriptive information of soil samples (unit: mg/kg).

| Element | Mean | Median | Standard Deviation | Minimum | Maximum | Coefficient of Variation | Background Value of Soil in Yunnan Province |
|---------|------|--------|-------------------|---------|---------|--------------------------|---------------------------------------------|
| Cd | 19.8806 | 12.7938 | 16.3516 | 2.9005 | 69.4155 | 82.25% | 0.66 |

#### 3.1.2. Descriptive Analysis of Spectral Curves

Using the spectral preprocessing method in 2.3.1, eight (RS, CT, SNV, MSC, D1, D2, DT, and CR) and nine (RS, CT, SNV, MSC, D1, D2, DT, CR, and CL) spectral transformations were applied to the XRF and vis-NIR spectra, respectively, to obtain seventeen post-transformation spectra, as shown in Table S1, Figures S2 and S3.

The transformed XRF and vis-NIR spectra exhibited significant differences in value ranges. The vis-NIR spectra had a maximum value of 56.48 and a minimum value of −29.15, whereas the XRF spectra had a maximum value of 859.00 and a minimum value of −137.46. This substantial difference in value ranges posed a challenge in directly concatenating the two spectra, as it would increase the difficulty in identifying fused spectral features. However, this disparity can be mitigated through spectral normalization techniques, which can eliminate such differences and facilitate feature selection using both PCC and CARS. The wavelength range for the XRF spectra was 0.39 to 53.205, while the vis-NIR spectra ranged from 350 to 2500. The number of spectral bands in the XRF and vis-NIR spectra was 3522 and 768, respectively. Hence, during PCC-based feature selection, it was advisable to increase the number of iterations for XRF spectrum selection to retain a greater amount of spectral information and ensure the interpretability of the data. The non-overlapping spectral ranges of the two types of spectra do not impact the uniqueness of the model variables.

### *3.2. Feature Spectrum Selection*

#### 3.2.1. Preliminary Feature Spectrum Selection Based on PCC

The PCC analysis between the spectral data and Cd concentration is depicted in Figure S4. It was observed that the correlation between the transformed vis-NIR spectra

and Cd concentration was enhanced. The highest correlation coefficient with Cd concentration was 0.47 for the D2 transformation. For the XRF spectra, most transformations improved their correlation with Cd, with the CR transformation showing the highest correlation coefficient of 0.51. The D1, D2, and CR transformations for both spectra exhibited correlation coefficients above 0.4, providing fundamental data for the establishment of a Cd concentration estimation model. However, relying solely on PCC for feature spectrum selection faces challenges in addressing inter-variable correlations. Building a Cd concentration estimation model based solely on PCC-selected feature spectra is prone to overfitting, resulting in unreliable Cd concentration estimations. Therefore, it was necessary to combine PCC with other methods for feature spectrum selection.

### 3.2.2. Feature Spectrum Selection Based on CARS Algorithm

The CARS algorithm primarily operates on the principle of "survival of the fittest" to select the feature variables with stronger adaptability, thereby eliminating irrelevant spectral information and retaining variables that can effectively improve the model's adaptability. Building upon the feature spectra obtained from the correlation analysis between soil Cd concentration and spectral values based on PCC, the CARS algorithm was applied to filter these feature spectra further. The number of iterations in the CARS algorithm was a crucial factor influencing the quality of the feature spectra. By setting the Monte Carlo sampling to 100 iterations, the relationship between the number of iterations and RMSECV is depicted in Figure S5. As the number of iterations increased, RMSECV exhibited fluctuating changes, with the curve initially decreasing, reaching a minimum point, and then rising again. The minimum RMSECV was achieved when the number of iterations was 79. Considering the combined impact of iteration count and computational time, setting the iteration count to 100 for the feature spectrum selection based on the CARS algorithm was deemed suitable.

### 3.3. Evaluation of Estimation Model Accuracy and Efficiency

#### 3.3.1. Evaluation of Single-Spectrum Estimation Models

From Table 2, it can be observed that vis-NIR exhibited lower accuracy in estimating Cd concentration. The Cd concentration estimation models established using nine spectral transformation methods had an average $R^2$ of 0.3529 and an average RMSE of 0.5388. By examining Figures 3 and S6, it is evident that most of the transformation models have large intercepts and small slopes, resulting in predicted values that are lower than the actual measurements. Among them, the trend lines of the D2 transformation model outperformed other models ($R^2$ = 0.6849, RMSE = 0.2690). Therefore, the D2 transformation model was selected as the optimal estimation model for soil Cd content in the vis-NIR range.

**Table 2.** Accuracy and efficiency statistics of 17 Cd concentration estimation models.

| Spectral Type | Transformation | RMSE | $R^2$ | Time (s) |
| --- | --- | --- | --- | --- |
| | RS | 0.7459 | 0.4054 | 18 |
| | CT | 0.4914 | 0.1123 | 339 |
| | SNV | 0.5288 | 0.2971 | 60 |
| | MSC | 0.5141 | 0.4402 | 30 |
| vis-NIR | D1 | 0.5065 | 0.3128 | 34 |
| | D2 | 0.2690 | 0.6849 | 110 |
| | DT | 0.5703 | 0.1405 | 51 |
| | CR | 0.5334 | 0.4781 | 14 |
| | CL | 0.6894 | 0.064 | 37 |

**Table 2.** *Cont.*

| Spectral Type | Transformation | RMSE | $R^2$ | Time (s) |
|---|---|---|---|---|
| XRF | RS | 0.3321 | 0.1082 | 71 |
| | CT | 0.4116 | 0.2628 | 59 |
| | SNV | 0.3018 | 0.7495 | 44 |
| | MSC | 0.4531 | 0.3841 | 71 |
| | D1 | 0.1143 | 0.9079 | 113 |
| | D2 | 0.1048 | 0.8868 | 164 |
| | DT | 0.3088 | 0.3772 | 16 |
| | CR | 0.1588 | 0.7442 | 68 |

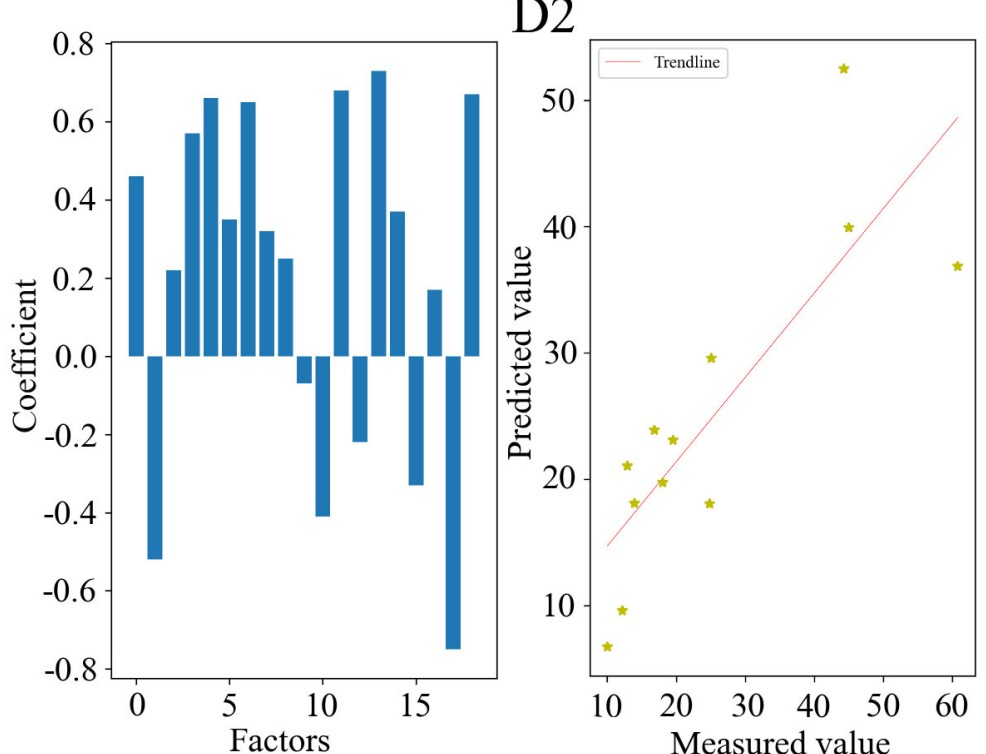

**Figure 3.** PLSR model with D2 transformation of vis-NIR (the rest of the transformations are in Figure S6).

The additional information in Table 2 indicates that XRF achieved better results in Cd concentration estimation. The excitation spectra of XRF can easily identify heavy metal elements in the soil. By examining Figures 4 and S7, it can be observed that the Cd estimation models based on XRF data had a higher number of spectral bands compared to the vis-NIR estimation models (Figures 3 and S6). The closer the scatter points of the XRF estimation models were to the 1:1 line, the higher the degree of fit of the Cd concentration estimation model. Among them, the scatter plots of the Cd concentration estimation models based on the SNV, D1, D2, and CR transformations exhibited smaller deviations from the 1:1 line compared to the other four methods (CT, MSC, DT, and CL), indicating better fit and higher estimation accuracy, with $R^2$ values of 0.7495, 0.9079, 0.8868, and 0.7442, and RMSE values of 0.3018, 0.1143, 0.1048, and 0.1588, respectively. Therefore, the SNV, D1, D2, and CR methods can be considered advantageous spectral transformation approaches for XRF spectra, providing a basis for further spectral fusion studies.

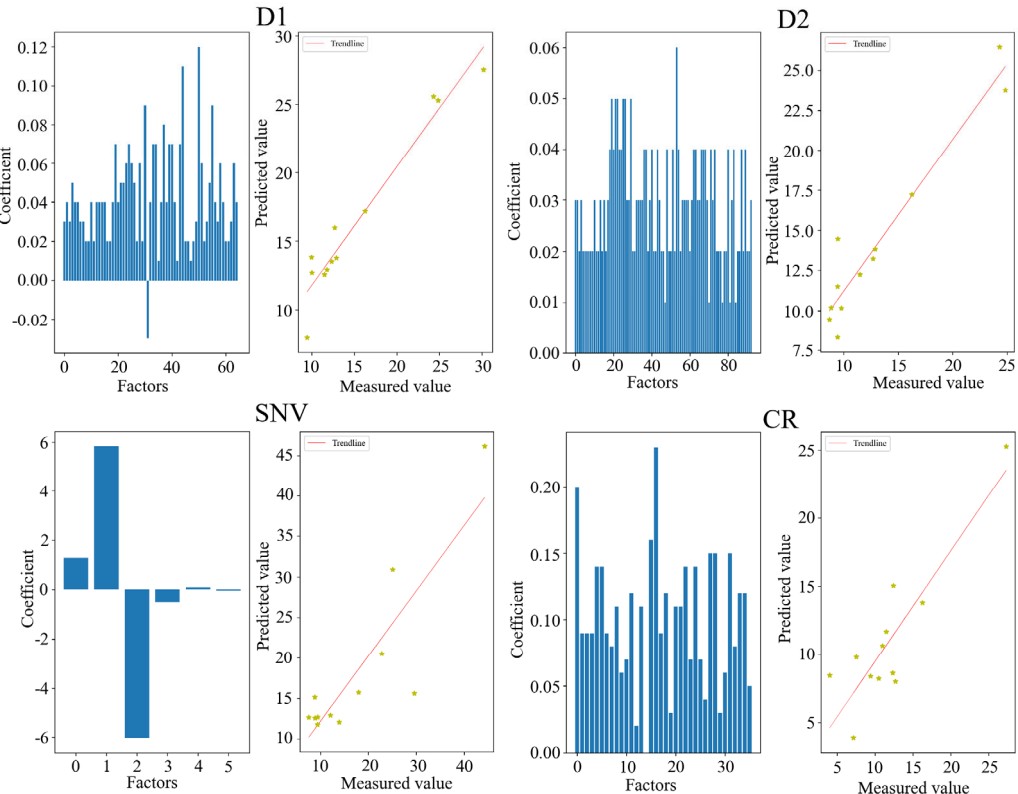

**Figure 4.** PLSR model with different transformations of XRF (the remaining transformations are in the attached Figure S7).

### 3.3.2. Evaluation of Spectral Fusion Models

Compared to the single-spectrum models, the fusion models exhibited higher accuracy. Table 3 presents the accuracy and computational time information of four fusion models: the concatenation of the D2 transformation of vis-NIR and the D1 transformation of XRF (D2(vis-NIR) + D1(XRF)); the concatenation of the D2 transformation of vis-NIR and the D2 transformation of XRF (D2(vis-NIR) + D2(XRF)); the concatenation of the D2 transformation of vis-NIR and the SNV transformation of XRF (D2(vis-NIR) + SNV(XRF)); and the concatenation of the D2 transformation of vis-NIR and the CR transformation of XRF (D2(vis-NIR) + CR(XRF)). Their $R^2$ values were 0.9505, 0.8832, 0.8974, and 0.9096, respectively, while the RMSE was 0.1174, 0.1486, 0.1904, and 0.1309, respectively. It is worth noting that although the fusion models had more spectral bands, their computational time was shorter compared to the single-spectrum models.

**Table 3.** Accuracy and efficiency statistics of four Cd concentration estimation models.

| Spectral Type | Transformation | RMSE | $R^2$ | Time (s) |
|---|---|---|---|---|
| FS | D2(vis-NIR) + D1(XRF) | 0.1174 | 0.9505 | 65 |
| | D2(vis-NIR) + D2(XRF) | 0.1486 | 0.8832 | 34 |
| | D2(vis-NIR) + SNV(XRF) | 0.1904 | 0.8974 | 18 |
| | D2(vis-NIR) + CR(XRF) | 0.1309 | 0.9096 | 23 |

The scatter plot in Figure 5 exhibits a linear distribution, indicating a good level of prediction. While the computational time of the fusion model D2(vis-NIR) + D1(XRF) was slightly longer, this fusion model demonstrates a linear and evenly distributed scatter plot, showcasing better prediction accuracy and stability. Moreover, it exhibited the least deviation from the 1:1 line, the highest degree of fit, and the best estimation model accuracy. Therefore, the D2(vis-NIR) + D1(XRF) fusion model can be regarded as the optimal estimation model for soil Cd content.

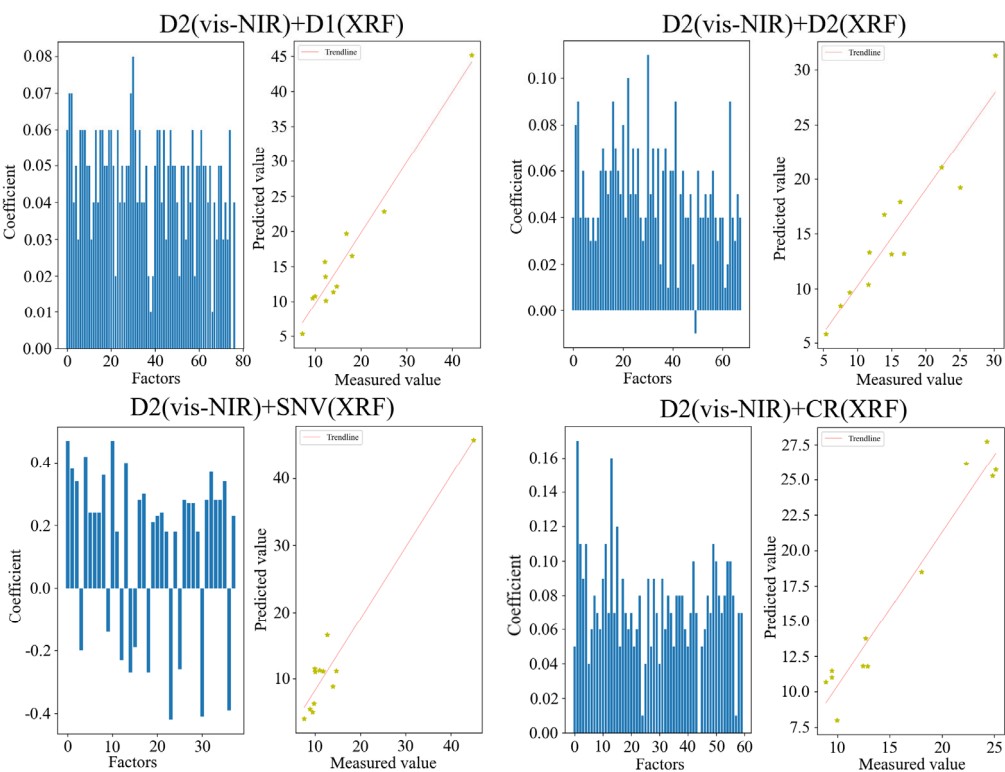

**Figure 5.** PLSR model with different transformation methods of FS.

Furthermore, Table 4 provides a comprehensive efficiency evaluation of the dominant transformation models for different spectral types, with mean estimates within the confidence interval ($p < 0.05$). Therefore, considering the average RMSE, average $R^2$, and average computational time, the ranking of estimation accuracy and efficiency for soil Cd concentration is as follows: the FS model was superior to the XRF model, and the XRF model was superior to the vis-NIR model. Taking D2(vis-NIR) as the reference, the average $R^2$ of XRF improved by 20.03%, while the average $R^2$ of the FS model improved by 32.90%. The average computational time of XRF decreased by 11.82%, while the average computational time of the FS model decreased by 68.19%. Compared with XRF spectroscopy, the average $R^2$ of FS was improved by 10.72%; the average operation time of FS was reduced by 63.92%. Overall, the FS model significantly improved the accuracy of the model, reduced algorithm runtime, and exhibited better stability.

**Table 4.** Evaluation table of the combined efficiency of the three models.

| Spectral Type | Transformation | Mean RMSE | Mean $R^2$ | Mean Time (s) |
|---|---|---|---|---|
| vis-NIR | D2 | 0.5334 | 0.6849 | 110 |
| XRF | D1, D2, SNV, CR | 0.1699 | 0.8221 | 97 |
| FS | D2(vis-NIR) + D1(XRF), D2(vis-NIR) + D2(XRF), D2(vis-NIR) + SNV(XRF), D2(vis-NIR) + CR(XRF) | 0.1468 | 0.9102 | 35 |

## 4. Discussion

Accurately establishing a model to estimate soil Cd concentrations is highly challenging in practical research. Currently, the primary method for monitoring soil Cd concentrations involves field technicians using specialized instruments to obtain Cd concentrations at certain sampling points, followed by spatial interpolation techniques to estimate Cd concentrations in specific areas. With the development of sensor devices, it is now possible to

measure an increasing amount of soil information non-destructively in the field. The development of machine learning has made it possible to analyze complex variable relationships using computers, providing more options for estimating soil Cd concentrations [18,21]. However, there are still numerous issues to be discussed regarding the estimation of Cd concentrations using spectroscopic techniques. Our discussion focuses on the following aspects.

First, as Cd is typically present in the form of compounds in soil, vis-NIR spectroscopy can indirectly estimate the concentration of Cd in soil by utilizing other substances that adsorb Cd, such as organic matter, carbonate minerals, clay minerals, and manganese iron oxides. For instance, Xia et al. [40] analyzed the correlation between eight soil heavy metals and organic matter and established a PLSR model for heavy metals. However, when the soil environment undergoes change, this method still requires on-site sampling to determine the concentrations of intermediate substances for quantitative inversion of heavy metal concentrations [41]. This effect prompted a better understanding of how to obtain information about in situ soil. In this regard, the XRF/NIR spectroscopy coupling technique proposed by Professor Horta expands the coverage of soil properties, improves measurement accuracy, and does not damage the original soil or generate hazardous laboratory wastewater [16]. Li et al. [18] combined XRF, NIR, and MIR spectroscopy and applied PCA to establish a model for estimating soil heavy metals. They found that the prediction model for Cd concentrations achieved an $R^2$ value as high as 0.98. In our study, a quantitative estimation model for soil Cd concentration was established by combining the serially linked vis-NIR and XRF spectra through the PCC_CARS_PLSR process. Through comprehensive comparison in Figure 6, it was found that the fusion model exhibited higher accuracy and shorter computation time, with the highest $R^2$ of 0.9896 achieved by the D2(vis-NIR) + D1(XRF) estimation model, indicating its greater potential for practical monitoring of soil Cd content.

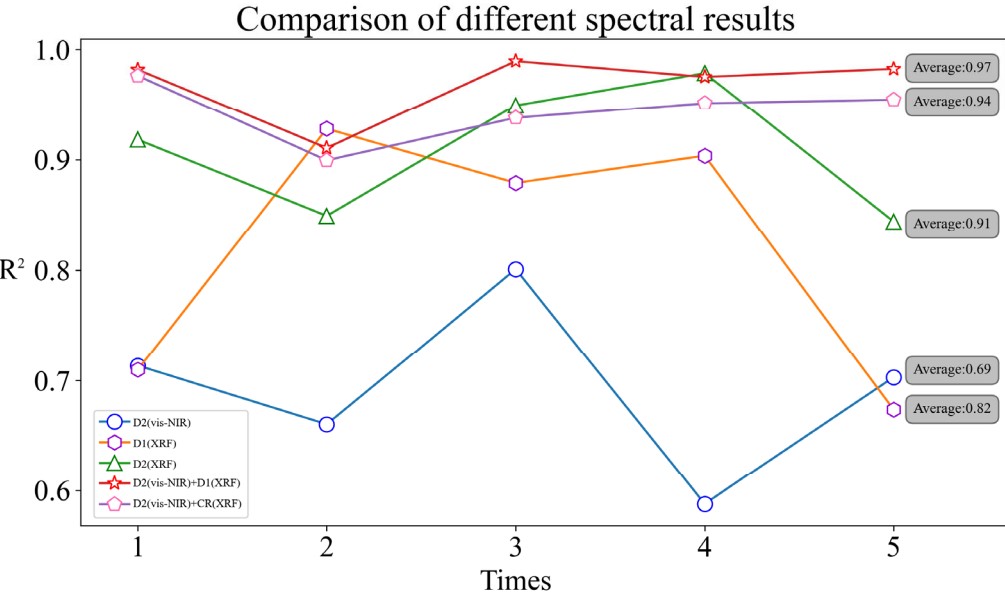

**Figure 6.** Comparison of $R^2$ for each of the 5 spectra with 5 operations.

Furthermore, soil spectral preprocessing was an effective approach to enhancing the accuracy of soil heavy metal estimation models. Spectral preprocessing typically involves spectral denoising and spectral transformation [42]. In our study, we utilized WT and spectral smoothing for spectral denoising. WT refined the received spectra from the sensor, reducing system noise. On the other hand, spectral smoothing employed a filtering window to minimize baseline drift in the spectral curves. Spectral transformation techniques enhanced spectral information, improving the correlation between spectra and soil Cd elements and providing additional valuable spectral information for feature spectrum

selection [43]. Hou et al. [44] achieved a high-accuracy estimation model by applying SG smoothing in combination with MSC and CL for preprocessing reflectance spectra. Therefore, employing a combination of multiple spectral preprocessing methods can provide more effective feature spectral information for high-accuracy estimation models. In our research, we combined the WT spectral denoising method with nine spectral transformations (CT, SNV, MSC, D1, D2, DT, CR, and CL). After spectral smoothing, a spectral variable matrix was obtained, and the accuracy of estimation models constructed using different preprocessing methods for individual spectra was evaluated. This evaluation served as a basis for selecting the optimal fusion method. Among the nine preprocessing methods applied to the vis-NIR spectra, the WT combined D2 preprocessing method yielded the highest accuracy with an $R^2$ of 0.6849. Similarly, among the eight preprocessing methods applied to the XRF spectra, the WT combined D1 preprocessing method yielded the highest accuracy with an $R^2$ of 0.9079. When compared with the optimal preprocessing methods for vis-NIR and XRF spectra in Gholizadeh et al. [45], our selected preprocessing methods demonstrated superior accuracy and stability.

In addition, spectral fusion can be categorized into three stages: data-level fusion; feature-level fusion; and decision-level fusion [16]. Data-level fusion, although straightforward, was limited by the effectiveness of data dimensionality reduction and feature spectral selection on the accuracy of heavy metal estimation models [46]. Feature-level fusion includes feature concatenation and feature operation [21]. Pozza et al. [20] combined PCA to concatenate the vis-NIR and XRF spectra, establishing a soil Pb concentration estimation model with a Lin's Concordance Correlation Coefficient (LCCC) of 0.95. Feature concatenation involves directly concatenating the selected feature spectra into a fused spectrum. Feature operation involved operations such as summation, equal-weight superposition, and outer-product analysis (OPA) on the vis-NIR and XRF feature spectra to obtain the fused spectrum [47]. Feature operation requires the dimensionality reduction in both the vis-NIR and XRF spectra to a unified number of spectral bands. Decision-level fusion typically employs Granger–Ramanathan averaging (GRA) to model the estimation results of individual spectral models. Xu et al. [46] used spectral resampling combined with OPA and GRA to estimate soil Cd concentration, resulting in LCCC values of 0.82 and 0.73, respectively.

We concatenated the vis-NIR and XRF spectra that underwent preliminary screening using PCC. Moreover, we employed CARS to identify the characteristic spectra, leading to the establishment of a quantitative estimation model for soil Cd concentration based on PLSR. This fusion of spectra can be categorized as feature-level fusion. Additionally, based on Table 2, various preprocessing techniques were applied to the individual spectra in this study. We used PCC_CARS integration to select the feature spectra, constructing a single-spectrum estimation model. Furthermore, on this basis, we selected the optimal fusion method for vis-NIR and XRF feature spectra, resulting in the final establishment of the fused spectral estimation model. This spectral fusion can also be categorized as decision-level fusion. Therefore, the fused spectral estimation model in this study is the outcome of multi-level fusion, with an average $R^2$ of above 0.94 for Cd concentration prediction in the validation set (Figure 6), which is similar to the accuracies of the soil Cd content estimation models developed by Xu et al. [46], Li et al. [18], and Wang et al. [21] The use of other spectral fusion strategies indicated that this multilayer fusion approach was a higher-order fusion strategy with higher accuracy.

In summary, this study fused the PCC-filtered vis-NIR and XRF spectra and employed the CARS method to identify characteristic spectra, resulting in the establishment of a quantitative estimation model for soil Cd concentration based on PLSR. The model demonstrated good prediction accuracy and stability. These findings provide theoretical guidance for non-destructive and rapid monitoring of soil Cd and enrich the methods for spectral techniques in monitoring soil Cd concentration. Moreover, they offer a technical approach for the on-site rapid estimation of soil Cd content based on multisource spectral fusion and lay the foundation for future research on dynamic, real-time, and large-scale quantita-

tive monitoring of soil heavy metal pollution using high-resolution hyperspectral remote sensing images.

## 5. Conclusions

In conclusion, the feature spectra were screened based on PCC-CARS coupling, and a soil Cd content estimation model with a higher integrated efficiency was developed using PLSR with the strategy of feature layer cascading and decision layer fusion. The comprehensive efficiency (accuracy and computational time) of the estimation models based on different spectral types was ranked as follows: fusion spectral model > XRF spectral model > vis-NIR spectral model. Among them, the D2(vis-NIR) + D1(XRF) fusion model exhibited the highest coefficient of determination and the smallest root mean square error. Importantly, compared with other spectral fusion methods, the concatenated spectral approach was characterized by simplicity and strong operability, which makes it more suitable for dynamic, real-time, and quantitative monitoring of soil heavy metal pollution. Future work should further focus on constructing models that can be used for on-site rapid and accurate estimation of soil Cd content, providing technical references for subsequent research on dynamic, real-time, and large-scale quantitative monitoring of soil heavy metal pollution based on high-resolution hyperspectral remote sensing images. This is of great significance for the dynamic monitoring of soil pollution and agricultural product safety, as well as the safe utilization of arable land.

**Supplementary Materials:** The following supporting information can be downloaded at: https://www.mdpi.com/article/10.3390/pr11092714/s1, Figure S1: Flow chart of modeling process. Figure S2: 9 transformations of vis-NIR spectra. Figure S3: XRF spectra with 8 transformations. Figure S4: Correlation coefficients between Cd concentration and spectral values. Figure S5: Relationship between Iterations and RMSECV. Figure S6: PLSR model for different transformation methods of vis-NIR. Figure S7: PLSR model with different transformation methods of XRF. Figure S8: Combined spectra of the two spectra. Table S1: Descriptive information of the spectral curves.

**Author Contributions:** Data curation, Z.Z., Y.L., J.Z. and D.T.; Funding acquisition, S.C.; Investigation, X.F., Q.Z. and C.L.; Methodology, Z.Z.; Project administration, Z.W. and Y.W.; Resources, Z.W. and Y.Z.; Software, Z.Z.; Supervision, Y.L.; Validation, L.C.; Writing—original draft, Z.Z.; Writing—review and editing, Z.W. All authors have read and agreed to the published version of the manuscript.

**Funding:** This work was supported by grants from the Ministry of Science and Technology of the People's Republic of China [grant numbers 2019YFC1803500, 2019YFC1803504]; the Natural Science Foundation of Sichuan Province [grant numbers 2018SZ0298, 2023YFS0390]; the National Natural Science Foundation of China [grant number No. 41402248]; Biological and Chemical Engineering Laboratory of Panzhihua College [grant numbers JDH-2019-E-01, GR-2020-E-02]; the Bureau of Science and Technology Panzhihua City [grant number 2017CY-N-8]; the Bureau of Science and Technology Aba Qiang Tibetan Autonomous Prefecture [grant number R22YYJSYJ0004, R23YYJSYJ0010]; the Southwest University of Science and Technology [grant numbers 17LZX613, 18LZX638].

**Data Availability Statement:** The data and materials used and analyzed during the current study are available from the corresponding author upon reasonable request.

**Acknowledgments:** The authors thank the Analytical Testing Center of Southwest University of Science and Technology for providing total heavy metal analysis services.

**Conflicts of Interest:** The authors declare no conflict of interest.

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
