# Peer review of "Quantitative Analysis of Soil Cd Content Based on the Fusion of Vis-NIR and XRF Spectral Data in the Impacted Area of a Metallurgical Slag Site in Gejiu, Yunnan"

_processes, doi:10.3390/pr11092714_

Round 1

Reviewer 1 Report

The manuscript revised are Quantitative analysis of soil Cd content based on the fusion of vis-NIR and XRF spectral data in the impacted area of a metallurgical slag site in Gejiu, Yunnan

In the introduction authors indicated that:

The study aims to combine XRF and vis-NIR spectra in a serial fusion approach, combined with PCC analysis, to construct  a technical roadmap for spectral preprocessing, feature spectrum selection, and quantitative estimation using the PCC_CARS_PLSR (Pearson Correlation Coefficient-Competitive Adaptive Reweighted Sampling-Partial Least Squares Regression) multi-method approach and they discussed and concluded well in the manuscript.

Authors need to clarify:

Among the 58 soil samples, 46 samples are selected for the training set, and 12 samples are selected for the validation set, how they selected the samples because they indicated that “The standard deviation of soil Cd is 16.3516, indicating significant differences in  Cd concentrations among different directions of non-ferrous metal smelting slag deposits.  Cd elements are influenced by topography, wind direction, and water sources, continu-280 ously flowing into surrounding areas”. Please explain the selection and number of the samples are important for the training and then the validation.

Authors need to specify and discuss other soil properties because vis-NIR and XRF results depends for example to soil texture and humidity, and  also the presence of other metals like Fe.  Also slag characterization seems to be important, it is the only source of contamination and clearly  indicated Why only Cd is studied?

Reviewer 2 Report

The manuscript is very interesting. However, it needs some minor reviews in order to be published. 

1. the map of sampling points should be added in the main text and not in the supplementary text. 

2. A geological map and a geological description of the area are a must. Please, improve the text. 

3. Please, improve the section on methodology. Did the author use standards? If yes, which one? What are the results? 

4. Improve the description for the set-ups in all instrumentation used. 

5. Could the author compare the obtained results with comparable areas? 

Minor errors. 

Reviewer 3 Report

1. Explain why u did quatitive analysis 

2. Explain why choose this model 

3. Where the problem statement 

Language need some enhancement 

Round 2

Reviewer 1 Report

The manuscript was substantially improved

Author Response

Please see the attachment. In order to distinguish the first modification, we use a blue mark for the content of the second modification.

Reviewer 2 Report

Apart from some minor English errors, the manuscript has reached the standard for publication.  

English needs verification on the next step. 

Author Response

Please refer to the attachment. In order to distinguish the first modification, we use blue markers as the content of the second modification.
